# Effects of Dietary Probiotic Supplementation on Egg Quality during Storage

**Camila Lopes Carvalho** [1], **Ines Andretta** [1,*], **Gabriela Miotto Galli** [1], **Thais Bastos Stefanello** [1], **Nathalia de Oliveira Telesca Camargo** [1], **Maiara Marchiori** [2], **Raquel Melchior** [1] and **Marcos Kipper** [3]

1. Departament of Animal Science, Faculdade de Agronomia, Universidade Federal do Rio Grande do Sul, Porto Alegre 91540000, Brazil
2. Postgraduate Program in Animal Science, State University of Santa Catarina (UDESC), Chapeco 89815630, Brazil
3. Elanco Animal Health, Sao Paulo 04703002, Brazil
* Correspondence: ines.adretta@ufrgs.br

**Abstract:** The objective of this study was to evaluate whether probiotic supplementation to laying hens can improve the quality of eggs during storage. The trial was developed in a commercial farm, in which light-weight laying hens (36 weeks old) housed in cages were randomly selected for one of two different treatments: a control group fed non-supplemented diets, or birds fed with diets supplemented with 50 g/ton of probiotics. The trial lasted for 84 days, comprising three productive phases of 28 days each. The fresh egg quality was evaluated and then the eggs were stored and randomly separated for quality assessment at each storage interval (7, 14, 21, 28, 35, and 42 days). Means were compared using variance analysis considering differences at 5 and 10%. The probiotic was able to improve albumen weight, yolk length, yolk height, and yolk index ($p < 0.05$) during storage. Yolk color (fan) was also improved by 3.9% ($p < 0.001$), while increases of 1.35% ($p < 0.001$) in luminosity, 8.05% ($p < 0.001$) in red intensity, and 3.4% ($p < 0.001$) in yellow intensity were observed in comparison to the control group. Probiotic treatment was able to reduce by 2.03% ($p < 0.001$) yolk pH, and by 19.65% ($p < 0.05$) TBARS levels when compared to the control treatment. Therefore, the addition of probiotics to laying hen diets is an effective strategy to improve egg quality during storage.

**Keywords:** dietary additives; egg characteristics; laying hen; shelf life

## 1. Introduction

Eggs are an excellent protein source, in addition to having vitamins, minerals (such as iron, phosphorus, selenium, and zinc), carotenoids, and essential fats. Eggs may also have antibacterial and antiviral properties by immune system modulation [1]. However, egg quality deterioration begins immediately after oviposition and keeps developing during storage, particularly in non-refrigerated environments. This deterioration is connected to several egg quality traits, such as albumen and yolk weight and pH. Losing eggs is a problem for food security worldwide and represents an important problem in the poultry industry [2]. For that reason, it is important to establish the shelf life of eggs and ensure food quality and safety for the consumer.

Pathogens that affect hens can also impair egg quality [3]. Thus, the feeding practice applied to birds is an effective way to modulate egg characteristics, since treatment can cause changes in flavor, freshness, and palatability [4]. The use of food additives is also one of the ways to modulate egg quality, and use of probiotics is an available tool.

Probiotics are live microorganisms with great potential to replace growth promoters that have been restricted in several countries [5] due to the inappropriate use of antibiotics both in human medicine and in animal production. The use of antibiotics is less frequent

in egg farming compared to other animal production activities. However, the benefits attributed to probiotics are still very important to laying hens.

These additives can increase the digestibility of protein and energy, and provide better animal performance, intestinal integrity, microbial profile [6], and immune system activity [7]. Probiotics also have an anti-inflammatory activity [8] and can increase short-chain volatile fatty acids, which are energy sources [6].

Different mechanisms of action have been attributed to probiotics. Biological effects (anaerobic bacteria contained in probiotics promoting an environment of low oxygen tension and thus inhibiting the growth of pathogens), chemical effects (production of bacteriocins), nutritional effects (competition for nutrients between beneficial bacteria of the probiotic and the pathogens), and physical effects (competitive exclusion or competition for a binding site) have been described [9–12].

Previous studies reported improvements in egg quality when birds were fed with probiotics [13,14]. However, to our knowledge, there are no studies on probiotic effects on egg shelf-life. For that reason, this study was developed to evaluate whether probiotic supplementation for laying hens impacts egg quality during different storage periods.

## 2. Materials and Methods

### 2.1. Animal Housing and Experimental Design

The project was developed on a commercial farm (Salvador do Sul, Rio Grande do Sul, Brazil) with around 14 thousand light-weight laying hens (Hyline W 36 lineage, 36 weeks old). Forty cages (with four laying hens each) were randomly selected on the farm and identified as part of the research project.

These replicates were assigned in a completely randomized design to the two treatments: the control (CON) treatment, which involved a basal diet, without supplementation with any other additive, and a probiotic (PRO) treatment involving the control diet supplemented with 50 g/ton of a multi-strain probiotic additive. The probiotic additive (Protexin Concentrate™, Elanco Animal Health, São Paulo, Brazil) included *Lactobacillus acidophilus* ($2.06 \times 10^8$ UFC/g), *Lactobacillus bulgaricus* ($2.06 \times 10^8$ UFC/g), *Lactobacillus plantarum* ($1.26 \times 10^8$ UFC/g), *Lactobacillus rhamnosus* ($2.06 \times 10^8$ UFC/g), *Bifidobacterium bifidum* ($2.0 \times 10^8$ UFC/g), *Enterococcus faecium* ($6.46 \times 10^8$ UFC/g), and *Streptococcus thermophilus* ($4.10 \times 10^8$ UFC/g).

The basal diet was a corn-soybean meal-based feed formulated according to the nutritional requirements of the bird species [15]. Inert material (kaolin) was included in the basal feed to replace the probiotic additive. Feed and water were provided ad libitum throughout the experimental period using gutter feeders and nipple drinkers.

The birds were housed in conventional sheds, arranged in an east-west direction, with concrete floors and masonry walls complemented with wire mesh to the ceiling. Each shed was equipped with side curtains, which were managed according to weather conditions to provide thermal comfort. The average minimum and maximum temperature and air relative humidity values recorded were 18 and 36 °C, and 35.8 and 94.7%, respectively. The lighting regime was composed of 16 h of light and eight hours of dark per day. The birds remained in galvanized-wire cages (100-cm long × 40-cm wide × 45-cm high, four birds each, resulting in a floor area of 500 cm²/hen) throughout the experimental period.

The experiment (supplementation) lasted for 84 days. For evaluation purposes, this period was divided into three different phases (phase 1, 36–40 weeks; phase 2, 41–44 weeks; and phase 3, 45–48 weeks). Egg sampling was conducted on the last day of each phase, when 240 eggs were randomly collected (120 from each treatment).

### 2.2. Egg Quality Assessment

Fresh egg quality was evaluated at day 1, and then the other eggs were stored at room temperature (25 °C) and randomly separated for quality assessment at each storage interval (7, 14, 21, 28, 35, and 42 days). Fifteen eggs per treatment were evaluated weekly, except for

the determination of substances that react to thiobarbituric acid (TBARS), total solids, and shell characteristics, whose attributes are described later in this document.

Eggs were identified and weighed individually at weekly intervals during the storage period. The *weight loss* (%) of eggs during storage was calculated as described by Caner and Cansiz [16], using the following equation:

$$Weight\ loss\ \% = \frac{((Final\ weight) - (Initial\ weight))}{(Initial\ weight)} \times 100$$

The albumen height was estimated by the average of three measurements taken at different points on the albumen at a distance of 10 mm from the yolk using a digital caliper (TMX PD–150, Precision Drilling, Calgary, AB, Canada). The Haugh Unit (*HU*) was obtained through the equation proposed by Haugh [17].

$$UH = 100\ log\ [H - \frac{\sqrt{(30W^{0.37} - 100)}}{100} + 1.19]$$

where $H$ = thickness of albumen (mm) and $W$ = mass of the entire egg (g).

Yolk width and height (mm) were measured with a digital caliper (TMX PD–150). The yolk index was calculated using the equation:

$$Yolk\ index = \frac{Yolk\ height}{Yolk\ width}$$

Yolk color was determined using a Roche colorimetric fan (DSM Animal Nutrition & Health, São Paulo, Brazil), with a score ranging from 1 (light yellow) to 15 (reddish orange). Color space was also assessed individually. In this analysis, a portable spectrophotometer device (Delta Vista model 450G, Novo Hamburgo, Brazil) was used to determine colorimetric coordinates of luminosity (L*), red intensity (a*), and yellow intensity (b*). The device was calibrated each day before use.

After yolk and albumen separation, the dense and the fluid albumen were homogenized for 20 s, and then the pH was determined using a digital pH meter (Kasvi model k39-2014B, Paraná, Brazil) previously calibrated with buffer solutions of pH 4, 7, and 10. The pH of the yolk was determined using the same device (Kasvi model k39-2014B, Paraná, Brazil) by inserting the electrode into each yolk in two randomly chosen spots.

Specific gravity was obtained according to Hempe et al. [18]. This method is based on Archimedes' principle, in which the value of specific gravity was obtained using the equation:

$$Specific\ gravity = \frac{Egg\ weight}{Egg\ weight\ in\ water \times temperature\ corretion}$$

The technique of Giampietro et al. [19] was used for the determination of lipid oxidation. TBARS was assessed in a pool of three yolks per treatment for three storage periods (0, 21, and 42 days). The decomposition of lipid peroxides was measured using a spectrophotometer (532 nm). The 1,1,3,3 tetramethoxypropane (TMP) component was used as a TBARS standard, and the results were expressed in mg TMP/kg yolk.

The method to achieve total solid content was determined in albumen and yolk, according to Pomeranz and Meloan (1994) [20]. Five grams of albumen and yolk were weighed separately in previously dried porcelain crucibles. The albumen and yolk samples were kept in an oven at 60 °C for 12 h and weighed. then the samples were kept at 105 °C for 12 h and weighed again. Seven eggs from each treatment were evaluated at fortnightly intervals to determine total solids.

Shell percentage was obtained after shell separation, washing, drying, and weighing on days 0, 21, and 42.

### 2.3. Statistical Analyses

A completely randomized design was used in the study. Each egg was considered an experimental unit. Statistical procedures were performed using SAS statistical software (9.4, SAS Inst. Inc., Cary, NC, USA). The normality of the data was verified and, afterward, the data were submitted to variance analysis using PROC MIXED. Statistical models considered the effects of treatment (control and probiotic), experimental phase (36–40, 41–44, and 45–48 weeks), days of storage (7, 14, 21, 28, 35, and 42 days), and interactions. To simplify results presentation, a table was populated with the overall means and probabilities for all responses evaluated in the study. Means were further evaluated (separately by phase and evaluation day) based on any effect ($p < 0.10$) relevant to the objective of the project (i.e., the effect of treatment or its interaction with phase and/or day). The probability of treatment effect was obtained for each storage day of each experimental phase. Probabilities were then interpreted at 5 and 10% significance.

## 3. Results

### 3.1. General Traits

No difference was observed in egg weight, weight loss, and specific gravity between treatments (Table 1). However, an interaction 'treatment by phase' was observed for weight loss ($p < 0.001$). Despite no effects being attributed to the treatments in phase 2 (Table S1), birds fed diets with probiotics produced eggs that had lower weight loss in phases 1 and 3 after 42 days of storage. Eggs from supplemented birds showed a cumulative weight loss 11% lower ($p < 0.01$) than the control group in phase 1, while a 15% lower weight loss was observed in phase 3 ($p < 0.001$).

**Table 1.** Quality of eggs from laying hens fed diets supplemented with probiotic.

| Responses | Treatments [1] | | *p*-Value [2,3] | | | | | | |
|---|---|---|---|---|---|---|---|---|---|
| | CON | PRO | Treatment (T) | Day (D) | Phase (P) | T × D | T × P | P × D | T × P × D |
| **General traits** | | | | | | | | | |
| Weight (g) | 61.76 | 61.96 | 0.532 | 0.294 | <0.001 | 0.099 | 0.187 | 0.004 | 0.125 |
| Weight loss (g) | 1.87 | 1.88 | 0.933 | <0.001 | 0.672 | 0.989 | <0.001 | 0.014 | 0.156 |
| Spec. gravity (g/mL) | 1.005 | 1.005 | 0.959 | 0.070 | 0.342 | 1.000 | 0.897 | 0.901 | 1.000 |
| **Albumen traits** | | | | | | | | | |
| Height (mm) | 4.278 | 4.244 | 0.432 | <0.001 | <0.001 | 0.695 | 0.200 | <0.001 | 0.556 |
| Weight (g) | 33.91 | 34.78 | 0.002 | <0.001 | <0.001 | 0.370 | 0.001 | 0.011 | 0.669 |
| pH | 9.16 | 9.15 | 0.323 | <0.001 | <0.001 | 0.980 | 0.206 | <0.001 | 0.524 |
| Haugh unit | 56.97 | 56.48 | 0.355 | <0.001 | <0.001 | 0.048 | 0.871 | <0.001 | 0.766 |
| **Yolk traits** | | | | | | | | | |
| Height (mm) | 13.12 | 13.28 | 0.006 | <0.001 | <0.001 | 0.131 | 0.001 | <0.001 | 0.042 |
| Length (mm) | 46.41 | 45.73 | <0.001 | <0.001 | 0.002 | 0.006 | 0.001 | <0.001 | 0.001 |
| Index | 0.290 | 0.295 | 0.002 | <0.001 | <0.001 | 0.801 | <0.001 | <0.001 | 0.003 |
| Weight (g) | 16.87 | 16.73 | 0.182 | <0.001 | <0.001 | <0.001 | 0.381 | <0.001 | 0.275 |
| pH | 6.40 | 6.27 | <0.001 | <0.001 | <0.001 | 0.231 | 0.263 | <0.001 | 0.047 |
| **Yolk color** | | | | | | | | | |
| Color score | 5.65 | 5.87 | <0.001 | <0.001 | <0.001 | 0.088 | 0.173 | <0.001 | 0.033 |
| Lightness (L*) | 56.09 | 56.85 | <0.001 | <0.001 | <0.001 | 0.777 | 0.763 | <0.001 | 0.005 |
| Redness (a*) | 6.33 | 6.84 | <0.001 | <0.001 | <0.001 | 0.031 | 0.023 | <0.001 | 0.272 |
| Yellowness (b*) | 56.54 | 58.46 | <0.001 | 0.123 | <0.001 | 0.103 | 0.006 | <0.001 | 0.686 |
| **Shell traits** | | | | | | | | | |
| Weight (g) | 5.86 | 5.92 | 0.095 | 0.002 | <0.001 | 0.289 | <0.001 | 0.007 | 0.395 |

[1] Means do not correspond only to fresh eggs but represent the whole sample of fresh and stored eggs. CON: Control feed, PRO: feed containing probiotics. [2] Means do not correspond only to fresh egg evaluation but represent an overall value comprising fresh and stored eggs. Quality assessment was performed on the last day of each phase (phase 1, 36–40 weeks; phase 2, 41–44 weeks; and phase 3, 45–48 weeks). Eggs were stored, and fifteen eggs per treatment were evaluated weekly (7, 14, 21, 28, 35, and 42 days). [3] Responses with treatment effect ($p < 0.10$) are fully described in the following tables. Significant interactions with treatment ($p < 0.10$) are described in the supplementary materials.

### 3.2. Albumen Traits

There were no differences between treatments for Haugh unit, height, and pH. However, an interaction 'treatment by day' was observed for Haugh unit ($p = 0.048$; Table 1). Eggs from laying hens fed diets containing probiotics showed a tendency ($p < 0.10$) towards higher Haugh unit mainly at the end of the trial (day 42, Table S2).

Probiotics increased the albumen weight by an average of 2.6% compared to the control treatment ($p = 0.002$). An interaction 'treatment by phase' ($p = 0.001$) was also observed for this response once a higher albumen weight was observed for the probiotic treatment on days 7 ($p = 0.036$/Table 2), 14 ($p = 0.020$), 21 ($p = 0.009$), 35 ($p = 0.003$), and 42 ($p = 0.033$) in phase 2, compared to the control group.

**Table 2.** Albumen weight (g) of eggs from laying hens fed with probiotics depending on storage time.

| Treatments | Storage Period (Days) | | | | | | |
|---|---|---|---|---|---|---|---|
| | 1 | 7 | 14 | 21 | 28 | 35 | 42 |
| | Phase 1–36 to 40 weeks | | | | | | |
| Control | 35.75 | 32.80 | 34.40 | 32.57 | 32.50 | 31.06 | 31.30 |
| Probiotic | 36.44 | 33.48 | 33.03 | 33.23 | 33.71 | 32.80 | 32.00 |
| *p*-value [1] | 0.642 | 0.622 | 0.326 | 0.518 | 0.369 | 0.062 | 0.200 |
| SE [2] | 0.720 | 0.668 | 0.688 | 0.496 | 0.662 | 0.470 | 0.567 |
| | Phase 2–41 to 44 weeks | | | | | | |
| Control | 36.80 | 34.10 | 32.60 | 31.38 | 31.94 | 29.96 | 30.71 |
| Probiotic | 36.16 | 37.13 | 35.41 | 34.17 | 33.88 | 33.01 | 33.22 |
| *p*-value | 0.613 | 0.036 | 0.020 | 0.009 | 0.227 | 0.003 | 0.033 |
| SE | 0.623 | 0.732 | 0.619 | 0.558 | 0.793 | 0.545 | 0.592 |
| | Phase 3–45 to 48 weeks | | | | | | |
| Control | 37.90 | 36.52 | 36.88 | 38.56 | 36.07 | 34.49 | 33.99 |
| Probiotic | 36.30 | 36.58 | 37.29 | 36.77 | 37.18 | 34.86 | 33.12 |
| *p*-value | 0.173 | 0.960 | 0.739 | 0.132 | 0.445 | 0.775 | 0.518 |
| SE | 0.582 | 0.514 | 0.600 | 0.589 | 0.714 | 0.619 | 0.658 |

[1] Probability of treatment effect. [2] Standard error.

### 3.3. Yolk Traits

There was an increase of 1% in yolk height in the probiotic treatment when compared to the control ($p = 0.006$), as well as an interaction between treatment by phase ($p = 0.001$) and treatment by phase by day ($p = 0.042$). The treatment with probiotics showed higher yolk heights in phase 1 on days 28 ($p = 0.002$/Table 3) and 35 ($p < 0.001$) when compared to the control group. The same was observed in phase 2 on days 7 ($p = 0.013$), 35 ($p = 0.040$), and 42 ($p = 0.007$) of storage.

**Table 3.** Yolk height (mm) of eggs from laying hens fed with probiotics depending on storage time.

| Treatments | Storage Period (Days) | | | | | | |
|---|---|---|---|---|---|---|---|
| | 1 | 7 | 14 | 21 | 28 | 35 | 42 |
| | Phase 1–36 to 40 weeks | | | | | | |
| Control | 18.66 | 15.77 | 13.15 | 11.82 | 10.51 | 10.07 | 9.96 |
| Probiotic | 19.13 | 15.36 | 13.10 | 11.94 | 11.21 | 11.01 | 10.07 |
| *p*-value [1] | 0.147 | 0.184 | 0.873 | 0.633 | 0.002 | <0.001 | 0.776 |
| SE [2] | 0.161 | 0.152 | 0.167 | 0.119 | 0.121 | 0.140 | 0.176 |
| | Phase 2–41 to 44 weeks | | | | | | |
| Control | 17.68 | 15.50 | 13.47 | 12.33 | 10.86 | 10.88 | 9.64 |
| Probiotic | 17.94 | 16.27 | 13.78 | 12.23 | 10.83 | 11.39 | 10.49 |
| *p*-value | 0.215 | 0.013 | 0.306 | 0.610 | 0.900 | 0.040 | 0.007 |
| SE | 0.106 | 0.159 | 0.152 | 0.099 | 0.116 | 0.126 | 0.164 |
| | Phase 3–45 to 48 weeks | | | | | | |
| Control | 17.59 | 15.97 | 14.05 | 13.07 | 12.39 | 11.48 | 10.74 |
| Probiotic | 17.75 | 15.81 | 13.65 | 12.67 | 12.38 | 11.23 | 10.79 |
| *p*-value | 0.636 | 0.564 | 0.103 | 0.131 | 0.961 | 0.315 | 0.872 |
| SE | 0.163 | 0.131 | 0.123 | 0.133 | 0.122 | 0.118 | 0.148 |

[1] Probability of treatment effect. [2] Standard error.

There was a decrease of 1.5% in yolk length in the probiotic treatment when compared to control ($p < 0.001$). In addition, there was an interaction between all factors analyzed ($p < 0.05$). A lower yolk length in the group fed with probiotics compared to the control was observed in phase 1 on days 21 ($p = 0.005$/Table 4) and 35 ($p = 0.017$) of storage. The same occurred in phase 2 on days 7 ($p = 0.005$), 14 ($p = 0.008$), 21 ($p = 0.001$), and 28 ($p = 0.003$). However, on day 1 ($p = 0.032$) in phase 3, we observed values higher than the control.

**Table 4.** Yolk length of eggs (mm) from laying hens fed with probiotics depending on storage time.

| Treatments | Storage Period (Days) | | | | | | |
|---|---|---|---|---|---|---|---|
| | 1 | 7 | 14 | 21 | 28 | 35 | 42 |
| **Phase 1–36 to 40 weeks** | | | | | | | |
| **Control** | 40.63 | 43.49 | 45.02 | 49.84 | 47.91 | 50.00 | 49,29 |
| **Probiotic** | 41.32 | 43.50 | 44.48 | 45.93 | 46.71 | 47.15 | 50.56 |
| *p*-value [1] | 0.261 | 0.986 | 0.328 | 0.005 | 0.098 | 0.017 | 0.175 |
| SE [2] | 0.299 | 0.277 | 0.272 | 0.730 | 0.361 | 0.613 | 0.463 |
| **Phase 2–41 to 44 weeks** | | | | | | | |
| **Control** | 41.43 | 44.73 | 45.89 | 47.80 | 50.39 | 47.90 | 51.59 |
| **Probiotic** | 41.33 | 42.51 | 44.19 | 45.30 | 47.57 | 48.47 | 51.02 |
| *p*-value | 0.814 | 0.005 | 0.008 | 0.001 | 0.003 | 0.536 | 0.680 |
| SE | 0.196 | 0.410 | 0.332 | 0.420 | 0.504 | 0.450 | 0.665 |
| **Phase 3–45 to 48 weeks** | | | | | | | |
| **Control** | 39.94 | 42.03 | 46.63 | 46.08 | 46.08 | 48.06 | 50.00 |
| **Probiotic** | 41.10 | 42.76 | 45.93 | 46.18 | 47.16 | 48.39 | 48.80 |
| *p*-value | 0.032 | 0.208 | 0.378 | 0.872 | 0.271 | 0.790 | 0.206 |
| SE | 0.273 | 0.289 | 0.393 | 0.297 | 0.431 | 0.600 | 0.470 |

[1] Probability of treatment effect. [2] Standard error.

Consequently, there was an increase of 2% in yolk index in the probiotic treatment when compared to the control ($p = 0.002$), as well as an interaction treatment by phase ($p < 0.001$) and treatment by phase by day ($p = 0.003$). The probiotic treatment showed a higher yolk index compared to the control in phase 1 on days 21 ($p = 0.018$/Table 5), 28 ($p < 0.001$), and 35 ($p = 0.002$) of storage. The same occurred in phase 2 on days 7 ($p < 0.001$) and 14 ($p = 0.020$).

**Table 5.** Yolk index of eggs from laying hens fed with probiotics depending on storage time.

| Treatments | Storage Period (Days) | | | | | | |
|---|---|---|---|---|---|---|---|
| | 1 | 7 | 14 | 21 | 28 | 35 | 42 |
| **Phase 1–36 to 40 weeks** | | | | | | | |
| **Control** | 0.460 | 0.365 | 0.287 | 0.239 | 0.219 | 0.206 | 0.209 |
| **Probiotic** | 0.464 | 0.353 | 0.299 | 0.260 | 0.240 | 0.234 | 0.200 |
| *p*-value [1] | 0.663 | 0.129 | 0.225 | 0.018 | <0.001 | 0.002 | 0.412 |
| SE [2] | 0.004 | 0.003 | 0.005 | 0.004 | 0.003 | 0.004 | 0.005 |
| **Phase 2–41 to 44 weeks** | | | | | | | |
| **Control** | 0.429 | 0.347 | 0.293 | 0.260 | 0.214 | 0.228 | 0.186 |
| **Probiotic** | 0.433 | 0.383 | 0.312 | 0.270 | 0.228 | 0.236 | 0.205 |
| *p*-value | 0.600 | <0.00 | 0.020 | 0.190 | 0.074 | 0.240 | 0.064 |
| SE | 0.003 | 0.004 | 0.004 | 0.003 | 0.003 | 0.003 | 0.005 |
| **Phase 3–45 to 48 weeks** | | | | | | | |
| **Control** | 0.441 | 0.380 | 0.314 | 0.284 | 0.271 | 0.239 | 0.220 |
| **Probiotic** | 0.432 | 0.366 | 0.301 | 0.275 | 0.263 | 0.229 | 0.222 |
| *p*-value | 0.340 | 0.155 | 0.147 | 0.197 | 0.399 | 0.271 | 0.848 |
| SE | 0.004 | 0.004 | 0.004 | 0.003 | 0.004 | 0.004 | 0.003 |

[1] Probability of treatment effect. [2] Standard error.

There was no difference between treatments for yolk weight. However, there was an interaction treatment by day ($p < 0.001$) and phase by day ($p < 0.001$). Lower yolk weight was observed on days 21 ($p = 0.003$/Table S3) and 42 ($p = 0.027$) of phase 2. A reduction of 2% in yolk pH was also observed in the probiotic treatment when compared to the control ($p < 0.001$), as well as an interaction between treatment by phase by day ($p = 0.047$). Lower pH values were observed in the probiotic group compared to control in phase 1 on days 21 ($p = 0.040$/Table 6) and 42 ($p = 0.038$) of storage, as well as in phase 2 on days 1 ($p = 0.038$), 7 ($p = 0.012$), 14 ($p = 0.030$), and 28 ($p = 0.004$), and in phase 3, on day 7 ($p = 0.007$) of storage.

**Table 6.** Yolk pH of eggs from laying hens fed with probiotics depending on storage time.

| Treatments | Storage Period (Days) | | | | | | |
|---|---|---|---|---|---|---|---|
| | 1 | 7 | 14 | 21 | 28 | 35 | 42 |
| | Phase 1–36 to 40 weeks | | | | | | |
| Control | 6.22 | 6.62 | 6.47 | 6.82 | 6.50 | 6.65 | 6.75 |
| Probiotic | 6.14 | 6.25 | 6.41 | 6.34 | 6.44 | 6.59 | 6.52 |
| *p*-value [1] | 0.172 | 0.088 | 0.732 | 0.040 | 0.613 | 0.612 | 0.038 |
| SE [2] | 0.030 | 0.094 | 0.084 | 0.120 | 0.057 | 0.056 | 0.057 |
| | Phase 2–41 to 44 weeks | | | | | | |
| Control | 5.89 | 6.06 | 6.11 | 6.31 | 6.85 | 6.35 | 6.49 |
| Probiotic | 5.85 | 5.97 | 5.96 | 6.29 | 6.52 | 6.26 | 6.44 |
| *p*-value | 0.038 | 0.012 | 0.030 | 0.776 | 0.004 | 0.257 | 0.512 |
| SE | 0.011 | 0.018 | 0.034 | 0.031 | 0.059 | 0.035 | 0.038 |
| | Phase 3–45 to 48 weeks | | | | | | |
| Control | 6.01 | 6.19 | 6.07 | 6.24 | 6.37 | 6.62 | 6.63 |
| Probiotic | 5.99 | 6.05 | 6.08 | 6.22 | 6.38 | 6.49 | 6.49 |
| *p*-value | 0.735 | 0.007 | 0.846 | 0.691 | 0.951 | 0.322 | 0.090 |
| SE | 0.022 | 0.073 | 0.026 | 0.034 | 0.041 | 0.062 | 0.043 |

[1] Probability of treatment effect. [2] Standard error.

### 3.4. Yolk Color

There was a 4% increase in yolk fan color in the probiotic treatment when compared to control ($p < 0.001$), and there was also an interaction between phase by day ($p < 0.001$) and treatment by phase by day ($p = 0.033$). Higher values of yolk color in the probiotic treatment when compared to control were observed in phase 1 on days 14 ($p = 0.031$), 28 ($p = 0.001$), and 42 ($p = 0.006$) of storage (Table 7). The same occurred in phase 2 on days 1 ($p = 0.003$) and 42 ($p = 0.006$), and in phase 3 on days 21 ($p = 0.005$) and 42 ($p = 0.045$) of storage.

**Table 7.** Yolk color score (palette) of eggs from laying hens fed with probiotics depending on storage time.

| Treatments | Storage Period (Days) | | | | | | |
|---|---|---|---|---|---|---|---|
| | 1 | 7 | 14 | 21 | 28 | 35 | 42 |
| | Phase 1–36 to 40 weeks | | | | | | |
| Control | 5.73 | 4.87 | 5.42 | 6.36 | 5.93 | 6.38 | 6.18 |
| Probiotic | 5.73 | 4.80 | 6.00 | 6.20 | 6.60 | 6.64 | 6.82 |
| *p*-value [1] | 0.999 | 0.764 | 0.031 | 0.525 | 0.001 | 0.348 | 0.001 |
| SE [2] | 0.172 | 0.108 | 0.137 | 0.121 | 0.106 | 0.135 | 0.109 |
| | Phase 2–41 to 44 weeks | | | | | | |
| Control | 5.33 | 4.57 | 4.67 | 6.07 | 6.80 | 6.46 | 6.36 |
| Probiotic | 5.93 | 4.87 | 4.80 | 6.00 | 7.00 | 6.79 | 6.87 |
| *p*-value | 0.003 | 0.135 | 0.493 | 0.826 | 0.094 | 0.139 | 0.006 |
| SE | 0.104 | 0.098 | 0.095 | 0.158 | 0.059 | 0.109 | 0.095 |
| | Phase 3–45 to 48 weeks | | | | | | |
| Control | 5.73 | 5.20 | 5.00 | 5.15 | 5.33 | 5.75 | 5.50 |
| Probiotic | 5.64 | 5.00 | 4.93 | 5.67 | 5.46 | 5.57 | 5.86 |
| *p*-value | 0.614 | 0.120 | 0.591 | 0.005 | 0.194 | 0.482 | 0.045 |
| SE | 0.087 | 0.059 | 0.060 | 0.095 | 0.118 | 0.123 | 0.089 |

[1] Probability of treatment effect. [2] Standard error.

An increase of 1% in lightness was observed in the yolk from the probiotic treatment compared to the control ($p < 0.001$). There was an interaction treatment by phase by day ($p = 0.005$) for this response. Greater lightness was observed in treatment with probiotics compared to the control in phase 1 on days 28 ($p = 0.006$/Table 8) and 35 ($p = 0.010$) of storage. The same occurred in phase 2 on day 21 ($p = 0.005$) of storage, and in phase 3 on days 14 ($p = 0.047$) and 35 ($p = 0.015$) of storage.

**Table 8.** Yolk lightness (L* color) of eggs from laying hens fed with probiotics depending on storage time.

| Treatments | Storage Period (Days) | | | | | | |
|---|---|---|---|---|---|---|---|
| | 1 | 7 | 14 | 21 | 28 | 35 | 42 |
| | Phase 1–36 to 40 weeks | | | | | | |
| **Control** | 51.30 | 58.18 | 57.60 | 57.13 | 55.27 | 55.38 | 57.77 |
| **Probiotic** | 51.48 | 57.51 | 58.27 | 57.48 | 57.86 | 57.53 | 58.96 |
| *p*-value [1] | 0.849 | 0.300 | 0.408 | 0.512 | 0.006 | 0.010 | 0.092 |
| SE [2] | 0.451 | 0.320 | 0.397 | 0.262 | 0.491 | 0.436 | 0.353 |
| | Phase 2–41 to 44 weeks | | | | | | |
| **Control** | 50.99 | 55.77 | 57.87 | 56.62 | 58.25 | 58.73 | 59.07 |
| **Probiotic** | 52.06 | 56.45 | 58.54 | 58.93 | 58.31 | 58.89 | 59.10 |
| *p*-value | 0.079 | 0.343 | 0.284 | 0.005 | 0.870 | 0.750 | 0.948 |
| SE | 0.305 | 0.353 | 0.307 | 0.431 | 0.187 | 0.248 | 0.262 |
| | Phase 3–45 to 48 weeks | | | | | | |
| **Control** | 50.26 | 53.37 | 54.61 | 56.49 | 57.56 | 57.09 | 58.61 |
| **Probiotic** | 50.45 | 54.60 | 56.57 | 56.69 | 57.34 | 58.03 | 58.97 |
| *p*-value | 0.768 | 0.167 | 0.047 | 0.733 | 0.760 | 0.015 | 0.384 |
| SE | 0.322 | 0.441 | 0.496 | 0.291 | 0.342 | 0.200 | 0.200 |

[1] Probability of treatment effect. [2] Standard error.

An increase in the red intensity of 8.05% was observed in yolks from the probiotic treatment when compared to control ($p < 0.001$), as well as an interaction treatment by day ($p = 0.031$) and treatment by phase ($p = 0.023$). A higher red intensity in yolks from the probiotic treatment compared to control was found in phase 1 on day 35 ($p = 0.003$/Table 9). The same occurred in phase 2 on day 1 ($p = 0.002$), 7 ($p = 0.006$), 21 ($p = 0.039$), 28 ($p = 0.001$), and 42 ($p < 0.001$) of storage.

**Table 9.** Yolk redness (a* color) of eggs from laying hens fed with probiotics depending on storage time.

| Treatments | Storage Period (Days) | | | | | | |
|---|---|---|---|---|---|---|---|
| | 1 | 7 | 14 | 21 | 28 | 35 | 42 |
| | Phase 1–36 to 40 weeks | | | | | | |
| **Control** | 7.58 | 6.03 | 7.21 | 7.02 | 6.57 | 5.80 | 7.01 |
| **Probiotic** | 7.63 | 6.28 | 7.28 | 6.71 | 7.15 | 7.28 | 7.45 |
| *p*-value [1] | 0.837 | 0.572 | 0.860 | 0.534 | 0.102 | 0.003 | 0.203 |
| SE [2] | 0.137 | 0.212 | 0.182 | 0.241 | 0.177 | 0.264 | 0.170 |
| | Phase 2–41 to 44 weeks | | | | | | |
| **Control** | 6.98 | 6.17 | 5.94 | 6.29 | 5.74 | 6.23 | 5.60 |
| **Probiotic** | 7.97 | 7.16 | 5.85 | 7.03 | 7.06 | 6.89 | 6.87 |
| *p*-value | 0.002 | 0.006 | 0.617 | 0.039 | 0.001 | 0.081 | <0.001 |
| SE | 0.173 | 0.188 | 0.085 | 0.179 | 0.209 | 0.187 | 0.190 |
| | Phase 3–45 to 48 weeks | | | | | | |
| **Control** | 6.81 | 7.03 | 6.32 | 6.09 | 5.49 | 5.58 | 5.39 |
| **Probiotic** | 7.40 | 6.54 | 6.48 | 6.38 | 6.11 | 6.22 | 5.89 |
| *p*-value | 0.211 | 0.195 | 0.751 | 0.512 | 0.235 | 0.125 | 0.097 |
| SE | 0.233 | 0.186 | 0.237 | 0.217 | 0.255 | 0.209 | 0.150 |

[1] Probability of treatment effect. [2] Standard error.

There was an increase in the yellow intensity of 3.4% in yolks from the treatment with probiotic compared to control ($p < 0.001$). In the same way, interaction treatment by phase was obtained ($p = 0.006$). Probiotic treatment had greater intensity of yellow when compared to the control group in phase 1 on day 28 ($p = 0.006$/Table 10) of storage. The same occurred in phase 2 on days 1 ($p = 0.036$), 7 ($p = 0.015$), 21 ($p = 0.014$), 28 ($p = 0.001$), and 42 ($p = 0.016$) of storage, and in phase 3 on days 28 ($p = 0.026$) and 42 ($p = 0.036$) of storage.

**Table 10.** Yolk yellowness (b* color) of eggs from laying hens fed with probiotics depending on storage time.

| Treatments | Storage Period (Days) | | | | | | |
|---|---|---|---|---|---|---|---|
| | 1 | 7 | 14 | 21 | 28 | 35 | 42 |
| | Phase 1–36 to 40 weeks | | | | | | |
| **Control** | 6.05 | 5.98 | 6.02 | 5.96 | 5.63 | 5.53 | 5.55 |
| **Probiotic** | 5.96 | 6.00 | 5.98 | 5.96 | 5.98 | 5.61 | 5.77 |
| *p*-value [1] | 0.178 | 0.880 | 0.799 | 0.997 | 0.006 | 0.688 | 0.252 |
| SE [2] | 0.338 | 0.533 | 0.642 | 0.551 | 0.656 | 0.977 | 0.929 |
| | Phase 2–41 to 44 weeks | | | | | | |
| **Control** | 5.70 | 5.52 | 5.62 | 5.40 | 5.75 | 5.86 | 5.65 |
| **Probiotic** | 6.07 | 6.02 | 5.80 | 5.83 | 6.18 | 5.98 | 5.91 |
| *p*-value | 0.036 | 0.015 | 0.254 | 0.014 | 0.001 | 0.133 | 0.016 |
| SE | 0.900 | 1.070 | 0.778 | 0.885 | 0.693 | 0.411 | 0.552 |
| | Phase 3–45 to 48 weeks | | | | | | |
| **Control** | 5.47 | 5.63 | 5.62 | 5.62 | 5.21 | 5.55 | 5.41 |
| **Probiotic** | 5.64 | 5.64 | 5.71 | 5.66 | 5.68 | 5.74 | 5.65 |
| *p*-value | 0.275 | 0.976 | 0.658 | 0.801 | 0.026 | 0.219 | 0.036 |
| SE | 0.793 | 0.579 | 0.905 | 0.656 | 1.08 | 0.748 | 0.595 |

[1] Probability of treatment effect. [2] Standard error.

### 3.5. Shell Traits

Probiotics showed a tendency to increase the eggshell weight by 1% compared to the control ($p = 0.095$). There was also a treatment by period interaction ($p < 0.05$) with improvement in shell weight as a response to probiotic treatment in phase 2 (days 1, 7, 14, and 35 of storage; $p < 0.05$; Table S4) and a tendency in phase 1 (fresh eggs).

### 3.6. Lipid Peroxidation and Total Solids

Lipid peroxidation data are presented in Table 11. Lower levels of TBARS were observed in fresh eggs from hens fed with probiotics when compared to control treatment from phase 3 ($p < 0.001$), while a tendency towards reduction was found for fresh eggs from phases 1 and 2 ($p < 0.10$). In addition, it was observed that probiotic treatment tended to reduce TBARS levels in the first phase on days 21 and 42 ($p < 0.10$) of storage.

There was no significant difference between treatments for total solids for both albumen and yolk ($p > 0.05$; Table S5).

**Table 11.** Thiobarbituric acid reactive substances in eggs from laying hens fed probiotics.

| Treatments | Storage Period (Days) | | |
|---|---|---|---|
| | 1 | 21 | 42 |
| | Phase 1–36 to 40 weeks | | |
| **Control** | 4.43 | 2.98 | 3.55 |
| **Probiotic** | 4.09 | 2.53 | 0.22 |
| *p*-value [1] | 0.056 | 0.088 | 0.051 |
| SE [2] | 0.21 | 0.22 | 0.16 |

**Table 11.** *Cont.*

| Treatments | Storage Period (Days) | | |
|---|---|---|---|
| | **1** | **21** | **42** |
| | *Phase 2–41 to 44 weeks* | | |
| **Control** | 4.77 | 3.91 | 3.22 |
| **Probiotic** | 3.78 | 3.94 | 3.24 |
| ***p*-value** | 0.075 | 0.877 | 0.958 |
| **SE** | 0.28 | 0.13 | 0.16 |
| | *Phase 3–45 to 48 weeks* | | |
| **Control** | 4.58 | 2.55 | 3.31 |
| **Probiotic** | 3.68 | 2.50 | 3.46 |
| ***p*-value** | 0.001 | 0.782 | 0.554 |
| **SE** | 0.17 | 0.08 | 0.11 |

[1] Probability of treatment effect. [2] Standard error.

## 4. Discussion

The gut microbial community can play an important role in the host's health and performance. For that reason, the ability of probiotic additives to promote modulatory effects on gut microbiota has prompted increasing scientific interest in the last decades [10]. Although previous research in poultry has demonstrated positive effects of probiotics on several performance outcomes, most published studies have focused on broilers and few pieces of evidence supporting the benefits of probiotics use in diets for laying hens are available in the literature [13,14]. However, eggs are among the most versatile and nutritious foods, and practices that can improve their quality are very important for producers and consumers.

Albumen is characterized as a clear colloidal solution that contains protein and is produced by epithelial cells in the magnum [21]. Hence, albumen quality is a parameter that reflects egg freshness [22] and protein quality. Thus, the increase in albumen weight observed in the probiotic treatment is probably due to higher protein deposition in these eggs. This may have occurred due to beneficial modulation of the intestinal microbiota, which provided better health and, consequently, better digestion and absorption of nutrients. However, it is important to highlight that most of the knowledge available on poultry science has been generated in broilers, which have different gut microbiota to those of layers. Thus, more gut microbiota-related studies are needed to better understand the role of different microbial communities in the performance of laying hens and in egg quality.

Yolk index reflects the information of a fresh egg [23]. This considers the height and length of the yolk, so the greater height and smaller length found in the probiotic treatment is an indication of an egg in which the effects of storage were minimized. Therefore, an increase in yolk index may be related to the ability and functionality of hepatocytes to synthesize vitellogenin [24]. Vitellogenin is a protein that transports lipids from the liver to the growing oocytes that give rise to the yolk. However, the exact mechanism of the probiotic is not known. It may be linked to the synthesis of estradiol and, as a result, to an increase in hepatic estrogen receptors, which are responsible for the synthesis of this protein. Furthermore, the lower pH value in the yolk of eggs in the probiotic-supplemented treatment is a beneficial effect and may be related to the higher deposition of antioxidants in the yolk that delay lipid peroxidation [25]. This hypothesis is supported by the increase in yolk color due to carotenoids and xanthophylls that have antioxidant properties, and by the lower levels of TBARS. TBARS is one of the most common methods for quantification of malondialdehyde (MDA), which is one of the end products formed by the decomposition of certain lipid peroxidation products [26].

Increase in yolk color is a desirable factor for consumers. Thus, the increase in the intensity of yellow and red is beneficial and depends on the carotenoid content present in the diet [27]. Gul et al. [28] reported that yolk color is related to the amount of xanthophylls and the antioxidant activity of these pigments, such as carotene. Therefore, the greater

amount of these pigments may explain the increase in yolk color and the decrease in lipid peroxidation observed in the probiotic treatment. In this context, it is known that lipid peroxidation is an undesirable factor, as it can cause a rancid taste and reduce the nutritional and sensory quality of eggs. Our data are in agreement with Tang et al. [27] who also observed an increase in yolk color when layers were supplemented with *Bacillus subtilis*. The increase in lightness (L* color) of yolk in the treatment supplemented with probiotic may have occurred due to the lower amount of solute present inside the yolk, which causes water to leave the intracellular medium to the extracellular medium. During this process an increase in humidity occurs on the gem surface due to greater reflection of incident light. As mentioned previously, more studies need to be developed to further understand the relation between gut microbial community and yolk color because the underlying mechanisms of action are not fully understood.

Some responses were influenced by probiotic supplementation only in some phases or storage days (interactions). This results probably indicate that the additive is more likely to improve egg quality under certain conditions, e.g., periods in which the productivity is higher, which can be considered a challenge for the animal's physiology. However, this hypothesis needs to be further evaluated in future studies.

In addition, few studies link the intestinal health of laying hens with egg quality. To our knowledge, this is the first study that has evaluated the interaction of gut microbiota using probiotics with egg shelf-life. More information in this research area is crucial because several environmental and physiological traits promote different gut microbiota in layers than in broilers, usually with a more complex and richer community [29]. Therefore, from this study, it can be observed that probiotics can delay and mitigate the negative effects of storage, such as the loss of pigmentation, and yolk and albumen quality. Benefits were probably related to the effects of probiotics improving the ecosystem of the gut in layers by balancing many of the microbial genera and, consequently, promoting the intestinal health of laying hens.

## 5. Conclusions

The present study indicates that a multi-strain probiotic additive (*Lactobacillus acidophilus*, $2.06 \times 10^8$ UFC/g; *Lactobacillus bulgaricus*, $2.06 \times 10^8$ UFC/g; *Lactobacillus plantarum*, $1.26 \times 10^8$ UFC/g; *Lactobacillus rhamnosus*, $2.06 \times 10^8$ UFC/g; *Bifidobacterium bifidum*, $2.0 \times 10^8$ UFC/g; *Enterococcus faecium*, $6.46 \times 10^8$ UFC/g; *Streptococcus thermophilus*, $4.10 \times 10^8$ UFC/g; 50 g per ton of feed) can increase egg quality in light weight laying hens (36 weeks old). Probiotic supplementation was able to improve albumen weight, yolk pH, yolk length, yolk height, yolk index, and TBARS. Furthermore, probiotics were shown to be efficient in improving yolk scores, which is required by consumers. Future studies are needed to elucidate a better connection between this additive, microbiota, and egg quality.

**Supplementary Materials:** The following supporting information can be downloaded at: https://www.mdpi.com/article/10.3390/poultry1030016/s1. Table S1: Weight loss (g) of eggs from laying hens fed with probiotics depending on storage time. Table S2: Haugh unit of eggs from laying hens fed with probiotics depending on storage time. Table S3: Yolk weight (g) of eggs from laying hens fed with probiotics depending on storage time. Table S4: Shell weight (g) of eggs from laying hens fed with probiotics depending on storage time. Table S5: Total solids of eggs from laying hens fed probiotics depending on storage time.

**Author Contributions:** Conceptualization, C.L.C., I.A. and M.K.; methodology, C.L.C., I.A. and M.K., validation, C.L.C., I.A. and M.K.; formal analysis, C.L.C., I.A. and M.K.; investigation, C.L.C., G.M.G., T.B.S., N.d.O.T.C. and M.M.; resources, I.A. and M.K.; data curation, C.L.C. and I.A.; writing—original draft preparation, C.L.C.; writing—review and editing, C.L.C., I.A., G.M.G. and M.K.; visualization, I.A. and M.K.; supervision, I.A., R.M. and M.K.; project administration, I.A. and M.K.; funding acquisition, I.A. and M.K. All authors have read and agreed to the published version of the manuscript.

**Funding:** This research was partially funded by Conselho Nacional de Desenvolvimento Científico e Tecnológico (CNPq) and Coordenação de Aperfeiçoamento de Pessoal de Nível Superior (CAPES).

**Institutional Review Board Statement:** The animal study protocol was approved by the Institutional Ethics Committee on the Use of Animals (CEUA) of Universidade Federal do Rio Grande do Sul (protocol code 39783; 21 December 2020).

**Informed Consent Statement:** Not applicable.

**Data Availability Statement:** Not applicable.

**Acknowledgments:** We thank Granja Petry for the donation of the eggs, and Elanco Animal Health for the donation of products.

**Conflicts of Interest:** Kipper, M was employed by Elanco Animal Health, São Paulo, Brazil. The remaining authors declare that the research was conducted in the absence of any commercial or financial relationships that could be construed as a potential conflict of interest.

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
