# Peer review of "Effects of Dietary Probiotic Supplementation on Egg Quality during Storage"

_poultry, doi:10.3390/poultry1030016_

Round 1

Reviewer 1 Report

This study evaluated whether probiotic supplementation to laying 14 hens can improve the quality of eggs during storage, and found that probiotic supplementation improved the yolk color, and increased the lightness, and reduced the TBARS in egg. The language needs to be greatly edited based on its current form.

More detailed information needs to be included in the methods.

Discussion should be improved.

How the L, a,b were evaluated?

Line 14, evaluate the hypothesis that whether probiotic supplementation in diet of laying hens.

Line 22-23, lightness or luminosity.

Line 51, the digestibility of protein and gross energy in diets, additionally providing….

Line 68-70. The birds were firstly addressed, then how caged.

Line 70, how many replicates.

Line 81, delete both.

Line 91, lasted for

Line 94, how do the eggs sampled.

Line 136, solid content, what do the word mean.

Line 228, “interaction treatment” should be “interaction effects”.

Line 229, delete “when compared to control”.

Line 237, “yolks” should be “yolk”.

Line 291, delete “regards amino acids”.

Line 314, L represents luminosity or lightness.

Line 328, was able to improve yolk color score.

Author Response

Dear editor and reviewers,
We are grateful for the time spent in reviewing our manuscript and for the valuable feedbacks. We have reviewed the manuscript considering all the suggestions. Please find the comments in the attached file.
Best regards.

Reviewer 2 Report

The conclusion must be much more specific and clarified (for example: name and dosage of probiotic usage, the production phase of additive usage, ...)

Author Response

(The authors gave the same response as above.)

Reviewer 3 Report

This manuscript includes some interesting results on egg quality with probiotic feeding. However, the results are shown complicatedly and the discussion part is not considered well with results. In addition, some numbers of Table in the manuscript are incorrect. I hope the manuscript will be revised well again.  

Author Response

(The authors gave the same response as above.)

Reviewer 4 Report

The manuscript presented for review presents original research results that can make a significant contribution to the current state of knowledge in the field of the presented topic. According to the Reviewer, the Introduction indicates the state of current knowledge. Experimental design and experimental techniques are suitable for solving the specific objectives of the study. The results are presented in an unbiased, clear, concise and complete manner. The discussion is relevant and adequate to the full interpretation of the results. The results and discussion justify the conclusions drawn from the work. I submit the suggestions marked in the text of the work to its authors for consideration.

Author Response

(The authors gave the same response as above.)

Round 2

Reviewer 1 Report

The revised manuscript was greatly improved.

Author Response

Thank you for your attention and support!

Reviewer 3 Report

This manuscript contains some interesting results. However, more considerations regarding the results are necessary for publication. Please refer to an attached file.

Author Response

All suggestions were considered in the new manuscript version. Thank you very much the attention and support.
